# Visual-Based Spatial Coordinate Dominates Probabilistic Multisensory Inference in Macaque MST-d Disparity Encoding

**DOI:** 10.3390/brainsci12101387

**Published:** 2022-10-13

**Authors:** Jiawei Zhang, Mingyi Huang, Yong Gu, Aihua Chen, Yuguo Yu

**Affiliations:** 1Shanghai Artificial Intelligence Laboratory, Research Institute of Intelligent and Complex Systems and Institute of Science and Technology for Brain-Inspired Intelligence, State Key Laboratory of Medical Neurobiology and MOE Frontiers Center for Brain Science, Human Phenome Institute, Shanghai 200433, China; 2Institute of Neuroscience, CAS Center for Excellence in Brain Science and Intelligence Technology, Chinese Academy of Sciences, Shanghai 200031, China; 3Key Laboratory of Brain Functional Genomics (Ministry of Education), East China Normal University, 3663 Zhongshan Road N., Shanghai 200062, China

**Keywords:** sensory integration and separation, cortical processing hierarchy, MST-d, causal inference

## Abstract

Numerous studies have demonstrated that animal brains accurately infer whether multisensory stimuli are from a common source or separate sources. Previous work proposed that the multisensory neurons in the dorsal medial superior temporal area (MST-d) serve as integration or separation encoders determined by the tuning–response ratio. However, it remains unclear whether MST-d neurons mainly take a sense input as a spatial coordinate reference for carrying out multisensory integration or separation. Our experimental analysis shows that the preferred tuning response to visual input is generally larger than vestibular according to the Macaque MST-d neuronal recordings. This may be crucial to serving as the base of coordinate reference when the subject perceives moving direction information from two senses. By constructing a flexible Monte-Carlo probabilistic sampling (fMCS) model, we validate this hypothesis that the visual and vestibular cues are more likely to be integrated into a visual-based coordinate rather than vestibular. Furthermore, the property of the tuning gradient also affects decision-making regarding whether the cues should be integrated or not. To a dominant modality, an effective decision is produced by a steep response-tuning gradient of the corresponding neurons, while to a subordinate modality a steep tuning gradient produces a rigid decision with a significant bias to either integration or separation. This work proposes that the tuning response amplitude and tuning gradient jointly modulate which modality serves as the base coordinate for the reference frame and the direction change with which modality is decoded effectively.

## 1. Introduction

The primate brain frequently combines multisensory information from different sensory modalities, such as information of visual, vestibular, auditory, and haptic origin, to improve the perception of the external world. Specifically, both visual and vestibular information can be used to infer motion direction, and combining the two is crucial for discriminating self-motion and object motion in a real-world situation. To achieve such discrimination, the neural system integrates or separates multisensory inputs to attribute information from different modalities to either a common source or two independent sources [1,2]. Nevertheless, the mechanisms through which neurons implement multisensory integration and separation have long been debated [3,4,5,6,7,8,9,10,11]. Although extensive computational studies provide various perspectives, there is a general lack of evidence from physiological records.

On the behavioral level, many researchers have demonstrated that the human brain adopts a discrimination strategy similar to the Bayesian approach [12,13,14,15,16,17], which makes probabilistic inferences based on the disparity between modalities as well as the changes in cue reliability. By representing the reliability as a prior distribution, the numerical fitting of the Bayesian model provides convincing approximations to the decision probabilities on the behavioral level. However, suffering from a gap between empirical results and mathematical description, the Bayesian strategy faces the challenge of other non-Bayesian strategies such as fixed inference, and even non-causal strategies such as forced fusion [18]. The inference strategy of brain multisensory processing remains elusive, mainly due to the lack of physiological evidence and the resultant lack of parameter constraints.

Previous studies seek neuronal correlates of multisensory processing in the primate dorsal medial superior temporal area (MST-d), which was a classical multisensory area responding to both visual and vestibular modalities [19,20,21]. Gu et al. found that the subtype of MST-d neurons (neurons preferring congruent and opposite stimuli directions) showed tuning sensitivity change that was analogous to behavioral change [1]. The experiments were designed to present visual and vestibular stimuli from the same direction on the horizontal plane. Festch et al. introduced a small conflict between the two directions (±4°) and proposed that MST-d neurons could account for weighing cues according to their relative reliabilities. Nevertheless, the MST-d neurons generally have large receptive fields [22], which drew attention to the response property on the 360° horizontal plane. Rideaux et al. trained an artificial network to perform causal inference and showed that a feedforward network can estimate the visual-vestibular motion by reading out the activity from neurons tuned to congruent and opposite directions [23]. WH Zhang proposed that congruent neurons encode integration while opposite neurons encode separation from a vector perspective, which covered the inference case when visual and vestibular directions are opposite [8]. JW Zhang et al. seek a correlation between neuronal preference and response property and proposed that the response strength ratio of MST-d neurons to both modalities determined the neurons serving as integration or separation encoders in a decision trial [24] and triggered the emergence of congruent and opposite neurons. The decision trial was denoted as explicit inference by Acerbi et al. [18], deciding whether the sensory cues originate from the same origin or separate origins. Nevertheless, the strength ratio did not specify visual and vestibular modalities. It is previously reported that the MST-d neurons generally present more significant tuning to visual modality than vestibular [25], but little is known about how such response discrimination affects the multisensory inference at a behavioral level, or whether the two modalities contribute to the perception distinctively at all.

The contribution potentially determines whether one modality serves as the reference frame, based on which the perception coordinates about self-motion are set up. During integration, the reference frame also determines whether the perceived common direction approximates closer to the visual modality or vestibular modality.

In this study, we seek to combine the physiological recordings and computational models and explore the key topic: What is the functional difference between the visual and vestibular signals during multisensory processing in MST-d? How does the response distinction affect the decision-making of integration or separation? We adopt a biophysical model that decides whether the senses should be integrated or separated. The model utilizes physiological recording results and reveals the heterogeneity of visual and vestibular contributions within MST-d.

## 2. Methods

### 2.1. Subjects and Surgery

Two male rhesus monkeys (*Macaca mulatta*) served as subjects. The general procedures followed in this study have been described previously [26,27]. Each animal was outfitted with a circular molded plastic ring anchored to the skull with titanium T-bolts and dental acrylic. To monitor eye movements, a scleral search coil was implanted in each monkey. Animals were trained to fixate on a central target for fluid rewards using operant conditioning.

### 2.2. Vestibular and Visual Stimuli

A 6-degree-of-freedom motion platform (MOOG 6DOF2000E; Moog, East Aurora, NY, USA) was used to passively translate the animals along one of eight directions in the horizontal plane, spaced 45° apart. A tangent screen was affixed to the front surface of the field coil frame and visual stimuli were projected onto it by a three-chip digital light projector (Mirage 2000; Christie Digital Systems, Cypress, CA, USA). The screen measured 60 × 60 cm and was mounted 30 cm in front of the monkey, thus subtending ~90° × 90°. The visual stimuli simulated translational movement along the same eight directions through a three-dimensional cloud of stars. Each “star” was a triangle that measured 0.15 cm × 0.15 cm; the cloud measured 100 cm wide by 100 cm tall by 40 cm deep and had a star density of 0.01 per cm^3^. To provide stereoscopic cues, the cloud was rendered as a red-green anaglyph and viewed through custom red-green goggles. The optic flow field contained naturalistic cues mimicking the translation of the observer in the horizontal plane, including motion parallax, size variations, and binocular disparity.

### 2.3. Electrophysiological Recordings

We recorded action potentials extracellularly from both hemispheres in each of the two monkeys. For each recording session, a tungsten microelectrode was passed through a transdural guide tube and advanced using a micromanipulator. An amplifier, an eight-pole bandpass filter (400–5000 Hz), and a dual voltage-time window discriminator (BAK Electronics, Mount Airy, MD, USA) were used to isolate action potentials from single neurons. Action potential times and behavioral events were recorded with 1 ms accuracy by a computer. Eye coil signals were processed with a low-pass filter and sampled at 250 Hz.

Magnetic resonance imaging (MRI) scans and Caret software analyses along with physiological criteria were used to guide electrode penetration into the MST-d area [25]. Neurons were isolated while a large field of flickering dots was presented. In some experiments, we further advanced the electrode tip into the lower bank of the superior temporal sulcus to verify the presence of neurons with response characteristics typical of the MT [25]. Receptive field locations changed as expected across guide tube locations based on the known topography of the MT [25].

### 2.4. Experimental Protocol

We measured neural responses to eight heading directions evenly spaced every 45° in the horizontal plane. Neurons were tested under three experimental conditions. (1) In vestibular trials, the monkeys were required to maintain fixation on a central dot on an otherwise blank screen while being translated in one of the eight directions. (2) In visual trials, the monkeys were presented with optic flow simulating self-motion (in the same eight directions), while the platform remained stationary. (3) In bimodal trials, the monkeys experienced both translational motion and optic flow. We paired all eight vestibular headings with all eight visual headings for a total of 64 bimodal stimuli. Eight of these 64 combinations were strictly congruent, meaning that the visual and vestibular cues simulated the same heading. The remaining 56 cases had conflicting cue stimuli. This relative proportion of strictly congruent and conflicting stimuli was adopted to characterize the neuronal combination rule. Each translation followed a Gaussian velocity profile. It had a duration of 2 s, an amplitude of 13 cm, a peak velocity of 30 cm/s, and a peak acceleration of 0.1× *g* (981 cm/s^2^).

These three stimulus conditions were interleaved randomly along with blank trials, which included neither translation nor optic flow. Ideally, five repetitions of each unique stimulus were collected for a total of 405 trials. Experiments with fewer than three repetitions were excluded from the analysis. When isolation remained satisfactory, we ran additional blocks of trials with the coherence of the visual stimulus reduced to 50% and/or 25%. Motion coherence was lowered by randomly relocating a percentage of the dots on every subsequent video frame. For example, we randomly selected one-quarter of the dots in every frame at 25% coherence and updated their positions to new positions consistent with the simulated motion, while the other three-quarters of the dots were plotted at new, random locations within the 3D cloud. Each block of trials consisted of both unimodal and bimodal stimuli at the corresponding coherence level. When a cell was tested at multiple coherence levels, both unimodal vestibular tuning and unimodal visual tuning were independently assessed in each block.

Trials were initiated by displaying a 0.2° × 0.2° fixation target on the screen. The monkeys were required to fixate on the target for 200 ms before the stimulus was presented and to maintain fixation within a 3° × 3° window to receive a liquid reward. Trials in which the monkeys broke fixation were aborted and discarded.

### 2.5. Data Analysis

The neural responses were binned in 100-ms time windows. Mean neural responses were averaged from 5 trials, and the units of measurement were spikes per second. The outliers in the 5 trials were removed, and the mean response was averaged from the remaining 4 trials. Using MATLAB (MathWorks, Natick, MA, USA), we chose the window of 750 ms to 1250 ms to select valid data. We considered a neuron to have discriminative tuning properties to one specific stimulus modality if the maximum response was 5 spikes/s more than the minimum response of the same curve. Tuning curve symmetry was not considered. Neurons that failed to meet this requirement for either the visual or the vestibular unisensory condition were considered unisensory-tuned neurons or poorly tuned neurons and removed from further analysis. Then, we computed the response ratio based on the visual and vestibular unisensory tuning curves of that neuron in the same time window. A threshold of 1.6 was chosen to discriminate between balanced and imbalanced data. We analyzed the tuning curve in the time window from 900 ms to 1250 ms, which corresponds to the maximum movement speed and maximum neural response. The window parameters were first scanned and then selected comprehensively to show the discrimination of the integration probability $P_{int}$ between the balanced and imbalanced neurons and to be physiologically plausible.

The group Δ*θ* distribution was rectified by doubling the probability at 0° and 180°, while the probabilities in other directions remained the same. This procedure was followed because of the experimental protocol in which directions were binned in 45° intervals; a 0° preference encompassed preferences from −22.5° to +22.5°, and a 180° preference encompassed preferences from 167.5° to 202.5°. However, each of the other preference bins was represented twice because both sides were included (for example, a Δ*θ* of 45° encompassed directions from 22.5° to 67.5° and from −22.5° to −67.5°). To align the widths of the probability bins, the data counted at 0°, and 180° were included twice.

### 2.6. Multisensory Tuning Curves Averaging

The multisensory tuning curve denoted as $f_{bal}$ and $f_{imbal}$ in balanced and imbalanced groups each were direction and modality averaged. The single maximal response in the two-dimensional multisensory response grid is first selected as the maximal response in $f_{bal}$ and $f_{imbal}$ $\left(R_{mul}^{max}=\max\left(R\left(\theta_{vis,mul},\theta_{vis,mul}\right)\right)\right)$. The other responses were computed as the averaged response of those with the same relative distance ($∆\theta_{rel}$) with the maximal one. For simplicity, we considered $R\left(\theta_{vis,mul}+∆\theta_{rel},\theta_{vis,mul}+∆\theta_{rel}\right)$ as same distant as $R\left(\theta_{vis,mul}+∆\theta_{rel},\theta_{vis,mul}\right)$ and $\left(\theta_{vis,mul},\theta_{vis,mul}+∆\theta_{rel}\right)$, thus the averaging is square-wise centered by the maximal response.

### 2.7. Bayesian Modeling

We adopted Bayesian optimal inference strategy as proposed in [17]. Here we explain the theory briefly. The strategy determines whether the sensory measurements $X_{vis}$ and $X_{ves}$ originates from a common source (*C* = 1) or separate sources (*C* = 2). If there is a common source, draw a position *s* from a normal prior distribution $N\left(0,\sigma_{p}\right)$, which stands for a normal distribution with 0 mean and standard deviation $\sigma_{p}$. Since its a common source, we denote $S_{vis}=S_{ves}=S_{common}$. If there are two sources, then the position $S_{vis}$ and $S_{ves}$ are independent of $N\left(0,\sigma_{p}\right)$. We assumed that visual and vestibular signals are corrupted by gaussian noise with standard deviation $\sigma_{vis}$ and $\sigma_{ves}$. Thus, the sensory measurements attribute to $X_{vis}\sim N\left(S_{vis},\sigma_{vis}\right)$ and $X_{ves}\sim N\left(S_{ves},\sigma_{ves}\right)$. The noise is independent across modalities. Given the statistical parameters, the posterior likelihood of a common source is expressed as,
(1)$$p\left(C=1|X_{vis},X_{ves}\right)=\frac{p(X_{vis},X_{ves}|C=1)p\left(C=1\right)}{p\left(X_{vis},X_{ves}\right)}$$

According to Bayes theorem,
$$p\left(C=1|X_{vis},X_{ves}\right)=\frac{p(X_{vis},X_{ves}|C=1)p\left(C=1\right)}{p\left(X_{vis},X_{ves}|C=1\right)p_{common}+\left(X_{vis},X_{ves}|C=2\right)\left(1-p_{common}\right)}$$

To $p\left(X_{vis},X_{ves}|C=1\right)$,
(2)$$p\left(X_{vis},X_{ves}|C=1\right)=\int p\left(X_{vis},X_{ves}|s\right)p\left(s\right)ds$$

Based on conditional independence hypothesis,
(3)$$\int p\left(X_{vis},X_{ves}|s\right)p\left(s\right)ds=\int p\left(X_{vis}|s\right)p\left(X_{ves}|s\right)p\left(s\right)ds$$

All the factors are Gaussians, then we obtain the analytical solution:(4)$$\begin{array}{c} p\left(X_{vis},X_{ves}|C=1\right)=\frac{1}{2\pi \sqrt{\sigma_{vis}^{2}\sigma_{p}^{2}+\sigma_{ves}^{2}\sigma_{p}^{2}+\sigma_{vis}^{2}\sigma_{ves}^{2}}}\\ \exp[-\frac{1}{2}\frac{{\left(X_{vis}-X_{ves}\right)}^{2}\sigma_{p}^{2}+{\left(X_{ves}-\mu_{p}\right)}^{2}\sigma_{vis}^{2}+{\left(X_{vis}-\mu_{p}\right)}^{2}\sigma_{ves}^{2}}{\sigma_{vis}^{2}\sigma_{p}^{2}+\sigma_{ves}^{2}\sigma_{p}^{2}+\sigma_{vis}^{2}\sigma_{ves}^{2}}] \end{array}$$
where $\mu_{p}=0$ is the mean of prior.

Similarly, to $\;p\left(X_{vis},X_{ves}|C=2\right)$,
(5)$$p\left(X_{vis},X_{ves}|C=2\right)=\iint p\left(X_{vis},X_{ves}|S_{vis},S_{ves}\right)p\left(S_{vis},S_{ves}\right)dS_{vis}dS_{ves}$$

We can derive the analytical solution as,
(6)$$p\left(X_{vis},X_{ves}|C=2\right)=\frac{1}{2\pi \sqrt{\left(\sigma_{vis}^{2}+\sigma_{p}^{2}\right)\left(\sigma_{ves}^{2}+\sigma_{p}^{2}\right)}}\exp[-\frac{1}{2}(\frac{{\left(X_{vis}-\mu_{p}\right)}^{2}}{\sigma_{vis}^{2}+\sigma_{p}^{2}}+\frac{{\left(X_{ves}-\mu_{p}\right)}^{2}}{\sigma_{ves}^{2}+\sigma_{p}^{2}})]$$

To reach a binary decision (C=1 or C=2), we assume an optimal Bayesian observer that reports a common source (C=1) when $p\left(C=1|X_{vis},X_{ves}\right)>0.5$, and vice versa.

In summary, the generative Bayesian strategy is featured by free parameters $\sigma_{p}$, $\sigma_{vis}$, $\sigma_{ves}$ and the prior $p_{common}$. We assume that $\sigma_{vis}=\sigma_{ves}$, reducing the free parameters to 3. The external cue disparity condition is simulated by the hyper-parameter $0{}^{\circ}\leq \left|S_{vis}-S_{ves}\right|\leq 180{}^{\circ}$. The fitting of fMCS model follows the least-squares estimate (LSE).

## 3. Results

### 3.1. Quantification of MST-d Neuronal Reliability Weightings Based on Tuning Curves

We began with experimental data analysis. The dataset includes physiological recordings of 158 MST-d neurons (as in [24]). In the experiments, monkeys were trained to actively perceive visual and vestibular stimuli with various directions (45° intervals) on the horizontal plane but were not required to give behavioral reports. Each neuron was recorded in three conditions: the visual and vestibular unisensory stimulus conditions and the visual-and-vestibular multisensory stimuli condition. Previous work proposed that the response balance between the two unisensory conditions determined neuronal function during the inference [24]. In this work, we similarly categorize the dataset as balanced and imbalanced neurons are discriminated by a ratio threshold of 1.6, which specifies only the balance between the dominant and subordinate unisensory responses and does not determine the visual or vestibular origin. The ratio in this work slightly deviates from previous work (1.7 in [24]) in choosing a larger time window, rendering the neuronal activity more moderate,
(7)$$r=\frac{\max\left(R_{vis}^{max},R_{ves}^{max}\right)}{\min\left(R_{vis}^{max},R_{ves}^{max}\right)}$$
where r is the response ratio, and $R_{vis}^{max}$ and $R_{ves}^{max}$ are the maximal values of preferred tuning curves of MST neurons to the visual and vestibular uni-sensory inputs.
(8)$$R_{vis}^{max}=\max\left[f_{0}\left(\theta_{vis}\right)\right]$$
(9)$$R_{ves}^{max}=\max\left[f_{0}\left(\theta_{ves}\right)\right]$$
where $\theta_{vis}$ and $\theta_{ves}$ represent the visual and vestibular input directions respectively, and $f_{0}$ is the neuronal spatial response function. As defined, it is obvious that the balanced neurons respond to both modalities with approximately equal amplitude (1~1.6) and the imbalanced neurons are characterized by the dominance of one modality, i.e., r ≥ 1.6 (Figure 1A; note that the maximal responses are shifted to align at 0 degrees).

We further examined the multisensory condition and then compared the responses of these two types of neurons with the unisensory condition. When both stimuli are presented simultaneously as a multisensory condition, the response is a function of both the visual and vestibular directions,
(10)$$R_{multi}=f_{multi}\left(\theta_{vis},\theta_{ves}\right)$$

Intuitively, the responses formed a two-dimensional grid along the two modalities. To unify the dimension of the tuning curves, the multisensory responses were averaged based on the spatial relative distance from the single maximal response ($R_{multi}^{max}$), and the averaged multisensory curves are presented in Figure 1A (gold lines). It is clear that the multisensory condition generally enhanced the responses of the balanced neurons (*df* = 70, *p* = 5.30 × 10^−16^, paired *t*-test, mean response gain = 26%), but the response from the imbalanced neurons was less enhanced (*df* = 55, *p* = 0.018, paired *t*-test, mean response gain = 8%). 

Figure 1B demonstrates $\max\left(R_{vis}^{max},R_{ves}^{max}\right)$ versus $R_{multi}^{max}$ in each neuron (top: balanced neurons, bottom: imbalanced neurons). Neurons with visual and vestibular unisensory dominance are presented with different colors. In balanced neurons, the visual and vestibular response distributions were not discriminated in either condition, suggesting that the visual and vestibular modalities share equal weights or contributions (Uni-sensory: *p* = 0.79, *df* = 67. Multisensory: *p* = 0.70, *df* = 67, two-sample *t*-test). However, the imbalanced neurons showed clear discrimination: both the unisensory and multisensory maximal responses of visual-dominant neurons ($R_{vis}^{max}>R_{ves}^{max}$) exceeded those of vestibular-dominant neurons ($R_{ves}^{max}>R_{vis}^{max}$) (Unisensory: *p* = 0.0086, *df* = 52. Multisensory: *p* = 0.0017, df = 52, two-sample *t*-test). Figure 1C shows in the time domain that the visual-dominant neurons generally respond more strongly than the vestibular-dominant neurons after stimulus onset (time = 0 s), suggesting an overall dominance of the visual modality in MST-d imbalanced neurons. Assuming that the response amplitude indicates modality weight in the psychophysical process, the visual information thus has greater potential to outweigh the vestibular information during the process of neural decision-making and alter the decision-making to present a bias towards visual information. In the next section, we validate this hypothesis by simulating decision-making with a biophysical model.

### 3.2. Balanced and Imbalanced MST-d Neurons Comprise Encoding Bases

Our model is a functional Monte-Carlo sampling (fMCS) model, using a plausible biophysical process to determine whether to integrate or separate the visual and vestibular external cues based on the balanced and imbalanced neuronal responses. As presented in Figure 2A, the fMCS model is mainly composed of a sampling module and a decision module (one decision neuron). The sampling module randomly chooses $n_{bal}$ and $n_{imbal}$ neurons with a preference according to the probability distribution, which is observed from the data (Figure 2A left). Approximated to the data observations ($n_{bal}\;$= 71, $n_{imbal}\;$= 56), $n_{bal}:n_{imbal}$ in the MCS model is set to 9:7 (72:56). To find a minimal requirement for reaching effective decisions, we set the numbers as 9 and 7. In previous works, we discussed that when the ratio does not meet this standard (for example, 1:1 or 1:2), the prior will deviate from 0.5 [24].

The neuronal preference determines when the neuron responds maximally (the peak response shown at 0° in Figure 1A) and characterizes how the neuron responds to specific external inputs. For simplicity, only the preference for disparity ($∆\theta_{multi}^{pref}$) is considered, and the absolute direction is omitted.
(11)$$\left[\theta_{vis,multi}^{pref},\theta_{ves,multi}^{pref}\right]=\mathrm{argmax}\left(R_{multi}\right)$$
(12)$$∆\theta_{multi}^{pref}=\min(\left|\theta_{vis,multi}^{pref}-\theta_{ves,multi}^{pref}\right|,360{}^{\circ}-\left|\theta_{vis,multi}^{pref}-\theta_{ves,multi}^{pref}\right|)$$

By definition, $∆\theta_{multi}^{pref}$ ranges from 0° to 180°. The external inputs take the form of disparity as well, denoted as $∆\theta_{cue}\;\left(0{}^{\circ}\leq ∆\theta ’\leq 180{}^{\circ}\right)$. When $∆\theta_{cue}$ is unaligned with $∆\theta_{multi}^{pref}$, the neuron response is weaker than the maximal response. In this case, the response is a function of the relative disparity ∆θ′,
(13)$$∆\theta \prime =\left|∆\theta_{cue}-∆\theta_{multi}^{pref}\right|$$

By averaging the data, the response under specific inputs is further quantified by the averaged multisensory tuning curves $f_{bal}$ and $f_{imbal}$, which are also data derived (Figure 2B side panel, same as the gold lines in Figure 1A). $f_{bal}$ and $f_{imbal}$ are direction-averaged and modality-averaged.
(14)$$R_{bal}=f_{bal}\left(∆\theta \prime \right)$$
(15)$$R_{imbal}=f_{imbal}\left(∆\theta \prime \right)$$

For example, if $∆\theta_{cue}$ is 30°, the neurons with $∆\theta_{multi}^{pref}$=30° show the maximal response because $f(\left|∆\theta_{cue}-∆\theta_{multi}^{pref}\right|)=f\left(∆\theta \prime =0{}^{\circ}\right)=R_{multi}^{max}$. However, the neurons with $∆\theta_{multi}^{pref}$=180° have a low response because $f\left(∆\theta \prime =150{}^{\circ}\right)$, which assigns a low response according to the curve. Different neurons respond distinctively according to Equations (14) and (15) because of different preferences. Therefore, sampling preferences represent sampling the response of a particular MST-d neuron, which is attributed to the various neuronal activation thresholds in biophysical conditions.

After sampling, a hypothetical decision neuron (decision module) receives the summed response outputs of the balanced and imbalanced MST-d groups ($R_{bal}^{tot}$ and $R_{imbal}^{tot}$) and decodes the source nature (∅) by comparing response amplitudes. Note that the balanced neurons generally serve as the separation basis, while the imbalanced neurons generally serve as the integration basis [24]. The growing imbalance of the responses between the two modalities pushed the neuron to have a single representation of the external cues. If the balanced group responds more strongly, then the decision separates the modalities and attributes the inputs to separate sources ($\emptyset \left(\mathrm{common}\;\mathrm{source}\right)=0$). Otherwise, they are attributed to a common source ($\emptyset \left(\mathrm{common}\;\mathrm{source}\right)=1$). The decision-making process is described as follows.
(16)$$R_{bal}^{tot}=\sum_{i=1}^{n_{bal}}R_{bal,i}$$
(17)$$R_{imbal}^{tot}=\sum_{j=1}^{n_{imbal}}R_{imbal,j}$$
(18)$$\left\{\begin{array}{c} \emptyset \left(\mathrm{common}\;\mathrm{source}\right)=0\;if\;R_{bal}^{tot}>R_{imbal}^{tot}\\ \emptyset \left(\mathrm{common}\;\mathrm{source}\right)=1\;if\;R_{bal}^{tot}\leq R_{imbal}^{tot} \end{array}\right.$$

$R_{bal,i}$ denotes the response from the $i^{th}$ sampled balanced neuron (Equation (14)), and $R_{imbal,j}$ denotes the $j^{th}$ sampled imbalanced neuron (Equation (15)). In conclusion, the input to the fMCS model is the hyper-parameter $∆\theta_{cue}$ ($0{}^{\circ}\leq \;∆\theta_{cue}\leq 180{}^{\circ}$) representing the external cue disparity. The output of each decision trial is either integration or separation and the decisions are repeated for 100,000 trials to obtain the probability of reporting a common source ($p_{int}=n_{int}/\left(n_{int}+n_{sep}\right)$).

As presented in Figure 2B, the fMCS model generated decision functions with a reverse sigmoidal shape, which had a high similarity to the behavioral results [18], especially in that the decision was prone to integrate at a small cue disparity and separate at a large cue disparity. Critically, the decision in any cue disparity condition was not determined but probabilistic due to the sampling nature.

We compared the decision function with the classical Bayesian causal inference strategy [17]. The Bayesian strategy fit closely with the fMCS decisions and further revealed that the best-fit prior was 0.51. Since we stated previously that imbalanced neurons with visual dominance have a higher response than those with vestibular dominance, it is evident that visual and vestibular dominance contribute to decisions differently. In the form of an averaged curve (Figure 2C, side panel), the visual-dominant imbalanced neurons responded with 73.55 spikes/s on average, while the vestibular-dominant imbalanced neurons responded with 50.45 spikes/s. Compared with the averaged balanced neuronal response (52.61 spikes/s), the visual-dominant neurons had an advantage in triggering an integration decision, while the vestibular-dominant neurons were subordinate to the balanced neurons and were prone to resulting in a separation decision. This was further validated by the fMCS model, which showed that the integration probability (p) was biased toward 1 if the fMCS model only contained visual-dominant neurons in the imbalanced group (Figure 2C, top) but biased toward 0 with vestibular-dominant imbalanced neurons only (Figure 2C, bottom). The classical Bayesian strategy predicted that the former case would be matched with an integration prior of 0.83, while the latter would be matched with 0.34. 

### 3.3. Discriminated Multisensory Tuning of Visual and Vestibular Modalities

From the analysis above, we conclude that the response amplitude is crucial to the final decision. Despite the distinction between the visual and vestibular dominance of the imbalanced neurons, the response function of each neuron was modality-averaged in the previous section. However, in realistic multisensory tuning, this may not be the case. As proposed previously, multisensory tuning is a two-dimensional grid (exemplified in Figure 3A), one dimension of which is the visual input direction and the other of which is the vestibular direction. The tuning on both dimensions may have different slopes (gradients), which determine the neuronal response at non-preferred inputs. A steep tuning gradient indicates that the neuron is more sensitive to the change in one modality [1] and vice versa. To study the modality-specific gradients individually, we denote the multisensory visual tuning curve (Figure 3B, black solid line) as
(19)$$R_{vis,multi}=f_{multi}(\theta_{vis,multi}|\theta_{ves,multi}^{pref})$$
where $\theta_{ves,multi}^{pref}$ is the preferred vestibular direction correlated with multisensory maximal response $R_{multi}^{max}$ and $\theta_{vis,multi}$ is the varying visual direction. Similarly, the multisensory vestibular curve (Figure 3B, black dashed line) is denoted as
(20)$$R_{ves,multi}=f_{multi}(\theta_{ves,multi}|\theta_{vis,multi}^{pref})$$

For each neuron, the multisensory visual or vestibular tuning curves intersect at $R_{multi}^{max}$. The responses at $\theta_{multi}^{pref}\pm \theta$ (0°≤θ≤180°) are averaged to obtain symmetric curves. The modality-specific tuning curves are further averaged across neurons in the same group. Figure 3C shows that the neurons were tuned more steeply to the visual modality than the vestibular modality in the balanced group, but the tuning gradients were proximate in general. In the imbalanced group, the visual-dominant neurons had a steep tuning gradient to the visual modality, while the responses to the vestibular modality were slightly tuned (Figure 3D,E). This suggests that, when the vestibular input deviates from the preference, the response remains constrained by the visual input. We simulated decision-making with the averaged balanced neurons as before but replaced the visual-dominant imbalanced neuronal tuning curve with the visual curve in Figure 3E. This is equivalent to simulating an external disparity ($∆\theta_{cue}$) with a visual-direction change only. Figure 3F shows that the visual direction change altered the decision from integration to separation following the growing disparity with the vestibular direction. This suggests that the decoded tuning of the visual modality is effective in representing external cue changes. The classical Bayesian strategy determined the best-fit prior to be 0.64, which is relatively close to a flat prior. However, when replacing the tuning curve with the vestibular curve in Figure 3E, the decision is robust integration across the vestibular direction change, in which case the Bayesian prior is obviously 1. 

Now let us focus on the vestibular-dominant imbalanced neurons. Compared with the visual-dominant imbalanced neurons, vestibular dominance did not lead to significant sharp vestibular tuning and blunt visual tuning; instead, the two curves presented close tuning gradients similar to those of the balanced neurons (Figure 3G–H), except that the vestibular tuning curve was slightly steeper than that of visual tuning. The decisions were simulated by an fMCS model similar to visual-dominance cases. Surprisingly, we found that such dominance did not produce an advantageous decision decoding to a vestibular direction change (Figure 3I). Rather, more flexible decisions were made with visual direction change again. The Bayesian strategy captured both modality changes with priors smaller than 0.5 (visual change: prior = 0.363, vestibular change: prior = 0.360). In conclusion, the vestibular modality was subordinate to the visual modality in MST-d both at the reference frame level and the effective decoding level.

## 4. Discussion

This work demonstrated the computational role of distinct MST neuronal response properties to either visual or vestibular inputs in the decision by using a response-based computational framework. 

We chose the functional Monte-Carlo sampling method in the framework because the sampling of response provides a concrete neuronal implementation to represent the randomization of perceiving the external world state. It was proposed that the probability of neuronal spiking is indeed computing the posterior probability of perception through response sampling [28,29,30,31]. As the main encoding components, different probability distributions of preference were observed empirically but the explicit mathematical description of the distribution was not accessible. By enough repetitions, the fMCS model carries out random sampling to approximate the probability density function, thus estimating the decision inference probability of the neural network. In our fMCS model, we sampled the neuronal response rate to represent the randomization of neuron activation as well as the perceived multisensory state of this neuronal unit. 

The analysis from the Bayesian perspective predicted well the results from the fMCS model, indicating that the multisensory computation in the physical cortex implemented causal inference with a nearly flat prior. Decision-making with flat prior means that the mature MST-d circuit is neither biased towards integration nor separation, which is most efficient for the neural system to address unknown cues. This further indicated that matured MST-d balanced and imbalanced neurons reached an ‘equilibrium’ state in causal inference, which was achieved through the counterbalance between the neuron number and response amplitude, combined with the preference distributions. Since the fMCS model in this work utilized neuronal response data from matured monkeys, it characterized the local-cortical decision properties after synaptic learning. 

Concerning the response amplitude, we present in Figure 1 that the relationship between unisensory and multisensory response matched with the additive effect found in previous works [25,32], and the impaired additive effect with growing response imbalance was predicted by the divisive normalization theory [33]. The division normalization is mainly attributed to intra-cortical shunting inhibition. Therefore, such passive modulation may not be the main site of synaptic learning discussed here. The learning process of multisensory inference is more likely to locate at the synaptic learning along the hierarchy [24,34], such as the connections from unisensory area MT (middle temporal) and PIVC (parietal insular vestibular cortex) to multisensory area MST-d, instead of the synaptic learning within the MST-d circuit. In other words, the learning is projected to unisensory responses rather than multisensory ones. 

We further propose that effective decision decoding requires not only matching the maximal response amplitude, but also the response gradients. The latter is a measurement of response selectivity, which is crucial to shifting the decisions flexibly between integration and separation based on the external cue disparity. 

In Figure 3, we demonstrate that the visual-dominant neurons respond selectively to the visual direction, but unselectively to the vestibular direction. As a result, the vestibular direction change did not affect the final decision and such rigid decoding has limited information capacity for vestibular-cue information. In addition, in vestibular-dominant neurons, the dominance did not produce an advantageous decision decoding to the vestibular direction change. We reason that vestibular-dominant imbalanced neurons generally respond at lower rates (mean maximal response is 50.45 spikes/s) than visual-dominant neurons (73.56 spikes/s) and balanced neurons (52.63 spikes/s). In this case, the weak tuning of the visual modality in turn enhanced the overall responses and pulled the decision closer to the equilibrium state.

The results indicated that the visual and vestibular modalities were weighed differently in MST-d, potentially leading to a biased reference frame for local decision-making [35]. The modality-specific neuronal response is the dynamic origin of reference frame selection, so the two modalities are prone to integration into a visual frame due to visual-response dominance but less likely to integrate into a vestibular frame. Furthermore, effective decision encoding mostly emerged from a visual direction change, while a vestibular direction change produced either robust integration or robust separation, making the decision rigid and inflexible. The analysis is compatible with previous experimental findings that MST-d was dominated by the visual frame when performing sensory integration [35]. It was previously debated whether an accurate decision requires a common frame, intermediate, or even distinct frames [36,37]. This work proposes that, in MST-d, the visual and vestibular frames are pre-coded concurrently by visual and vestibular-dominant neurons. Due to the prevalence of visual signals in MST-d [25], the integration is mostly based on the visual frame. Nevertheless, we reason that the vestibular frame serves as a fundamental role during the decision, because it also provides a base coordinate about self-motion for the vestibular-related signal if the decision is to separate the cues. 

## 5. Conclusions

This study has combined the methods of computational modeling and analysis of physiological recordings and examined the effect of functional heterogeneity between visual and vestibular modalities in MST-d. The heterogeneity stemmed from distinct tuning selectivity, which denotes two aspects: the response amplitude, and the tuning gradient (the tuning curve’s steepness). Previously, related works mainly explored the multisensory causal inference on two levels: one is the mathematical principle of multisensory behaviors [12,17,18], and the other is biophysical computation modeling at a neuronal circuit level [1,2,8,33]. Nevertheless, few works have bridged the gap between the two levels and studied the decision property stemming from neuronal architecture. This work initialized from the neuronal response and cortical processing hierarchy and reproduced decisions with Bayesian strategy. Importantly, we reported that the modality which produces larger neuronal response amplitude and steeper tuning is more likely to have a larger weight in multisensory decisions. Furthermore, we predicted that the response dominance of one modality determined the base coordinate to set up the reference frame for motion detection. In conclusion, the neuronal network response patterns compose the internal representation of external motion relationships, which may serve as a ubiquitous probabilistic-inference base for the merging of senses.

While the present work focused on the primate MST-d network, past studies demonstrated that both human and primate behavioral properties exhibited the Bayesian inference approach [1,12,34]. Despite the lack of physiological evidence of human subjects, it was reported that human brain contains the homolog of macaque MST area, which locates in human MT+ complex and compiles both visual and vestibular signals [38,39,40]. As Zhang et al. proposed [24], the macaque MST-d provided essential components for Bayesian population coding, thus future works may need to examine whether human follows similar principle to make a multisensory inference.

## 6. Limitations

As we seek to provide a general principle for biophysical sampling-based processes, the present work suffers from several limitations. It has been demonstrated that real decision processes in the brain also involve sampling of neuronal responses [31]. In MST-d, the neuronal response shows complex modulation of preferences and inputs, and the response decay may not be symmetrical when the inputs deviate from the preference of each input modality. Furthermore, the preference from unisensory to multisensory conditions may change, potentially causing a sharper decrease in neural responses if one input is unreliable. Both conditions add more variability to the decision process, which induces more flexibility. A high-dimensional sampling process would be required to include this variability.

## Figures and Tables

**Figure 1 brainsci-12-01387-f001:**
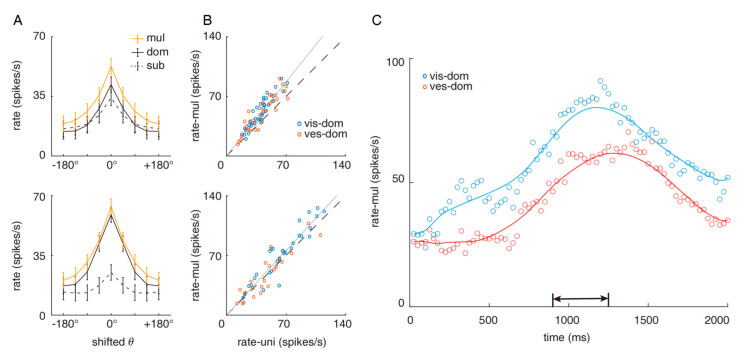
Visual and vestibular response distinction in balanced and imbalanced MST-d neurons. (**A**) Averaged tuning curves of balanced (top) and imbalanced (bottom) neurons in the monkey brain dorsal medial superior temporal area (MST-d). Black solid lines denote the curves corresponding to the modality of $max\left(R_{vis}^{max},R_{ves}^{max}\right)$, and black dashed lines denote the curves corresponding to the modality of $\min\left(R_{vis}^{max},R_{ves}^{max}\right)$. Gold solid lines denote the multisensory curve, which is direction-averaged and modality-averaged. Error bars denote the mean standard error. (**B**) Comparison of the unisensory and multisensory maximal fire-rate amplitudes of balanced (top) and imbalanced (bottom) neurons. Blue: visual-dominant ($R_{vis}^{max}>R_{ves}^{max}$). Red: vestibular-dominant ($R_{vis}^{max}<R_{ves}^{max}$). Each circle denotes a neuron. The dominance is determined by the unisensory responses. (**C**) Mean dynamics in the time domain for MST-d visual-dominant (blue) and vestibular-dominant (red) neurons. Each circle denotes the averaged response of corresponding neuron group at a particular time step. The area indicated by the two-headed arrow is the time window in which the averaged responses are computed.

**Figure 2 brainsci-12-01387-f002:**
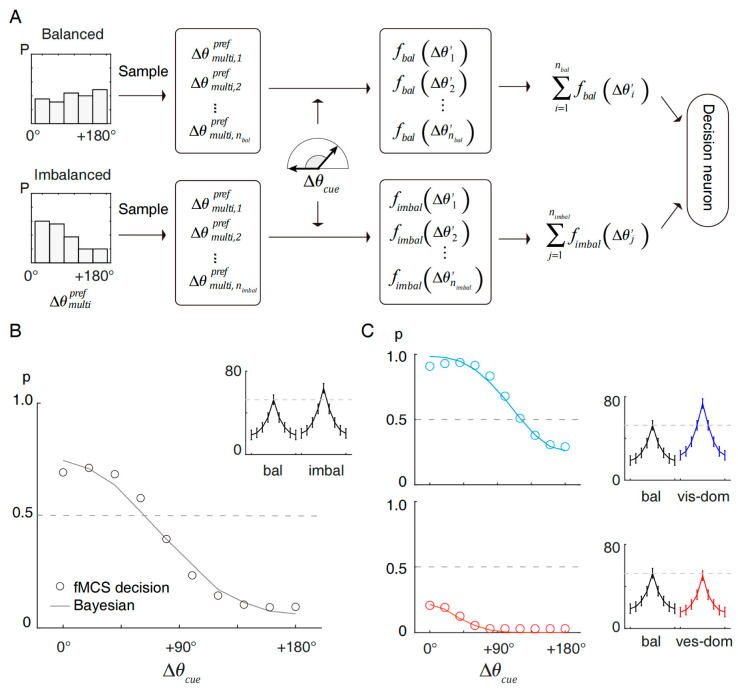
Functional Monte-Carlo sampling model that indicates the distinct roles of visual and vestibular neurons during decision-making. (**A**) Schematics of the MCS model for monkey MST-d. The model input is the stimulus disparity ($∆\theta_{cue}$). The neuronal preference is sampled from each distribution of balanced and imbalanced groups and transformed into the fire rate by a response function (f). The decision of integration or separation is reached by comparing the sampled group response from balanced and imbalanced neurons. (**B**) Probabilistic decision functions produced by the MCS model (circles) and fitted by the Bayesian model (solid curves). The decision probability p of whether to integrate the cues (reporting cues from the same source) is a function of the external cue disparity ($∆\theta_{cue}$). The side panel demonstrates the averaged response functions in the simulation, which are shared by all sampled neurons. (**C**) The decision by the fMCS model with visual-dominant imbalanced neurons only (top) or vestibular-dominant imbalanced neurons only (bottom). The visual-dominant neuronal response curves are presented in the top side panel (blue curve) and the vestibular-dominant neuronal response curves are presented in the bottom side panel (red curve).

**Figure 3 brainsci-12-01387-f003:**
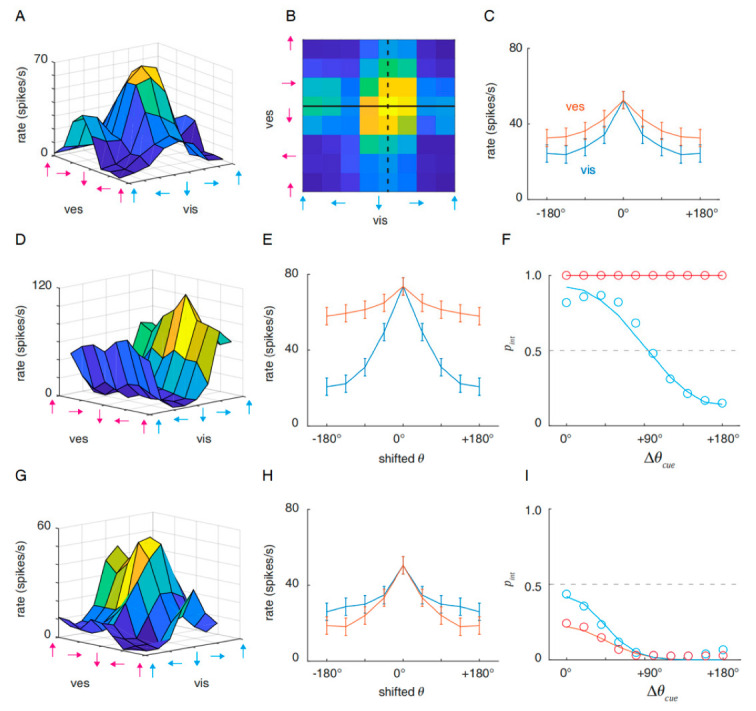
Distinct multisensory visual and vestibular tuning affects the decision. (**A**) Multisensory tuning grid for a balanced neuron example. (**B**) The 2-dimensional contour of the response grid in A. The black solid line denotes $f_{multi}(\theta_{vis,multi}|\theta_{ves,multi}^{pref})$, which is the multisensory visual curve, and the black dashed line denotes $f_{multi}(\theta_{ves,multi}|\theta_{vis,multi}^{pref})$, which is the multisensory vestibular curve. The two curves intersect at $R_{multi}^{max}$. (**C**) Visual (blue) and vestibular (red) multisensory response curves for balanced neurons, *n* = 71. The curves are averaged to be symmetric. (**D**) Multisensory tuning grid for a visual-dominant imbalanced neuron example. (**E**) Averaged multisensory curve of visual-dominant imbalanced neurons, *n* = 32. (**F**) The decision inference determined by the fMCS model, with the averaged balanced curve as in Figure 2 and the visual (blue) and vestibular (red) curves of the visual-dominant imbalanced neurons. (**G**–**I**) Same as (**D**–**F**), for vestibular-dominant imbalanced neurons.

## Data Availability

The datasets analyzed in this study are available from the corresponding author for reasonable request. The key analysis codes are available for reasonable request.

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
