# Peer review of "Visual-Based Spatial Coordinate Dominates Probabilistic Multisensory Inference in Macaque MST-d Disparity Encoding"

_brainsci, 2022, doi:10.3390/brainsci12101387_

Round 1

Reviewer 1 Report

I recommend major revision and resubmission.

1. The method and materials section is missing. That is not acceptable for an empirical study. I would request the authors to fill in the Method section part first. Even though the data they used in the modelling part are from a previous study (Zhang et al. 2022), it is necessary to have a methods section to describe the experimental procedure and materials briefly.

2. The title, abstract, and introduction need to specify whether the studies/theories/conclusions were based on animal studies or human studies. If there is evidence to suggest that humans and monkeys follow the same principles in multisensory integration, please specify in the introduction and discussion. The Introduction sections needs to be expanded on the literature review. What tasks and subjects had been used in the previous studies? Did they use the same experimental protocol in testing sensory integration in the MST-d area? For example, what is a
decision trial? Exactly what were the controversial points from previous studies? Does the experimental task matter? Does learning experience matter? Without a proper context detailing what the previous work had done, it is difficult to appreciate the significance of the research questions in this study. Since this is a modelling study, what did previous work do in this regard? What kind of models had been implemented? Why was functional Monte-Carlo sampling (fMCS) model chosen to address your research questions? The authors need to provide the rationale and background information in the Introduction.

3. The Results sections described how they collected the data using 6 lines in the first paragraph. This is not a section to describe the methods. Readers should not be required to find and read another paper to know the basic experimental procedure. For example, what were the monkeys asked to do in the visual and vestibular unisensory stimulus conditions and the visual-and-vestibular multisensory stimuli condition? How was the experimental design different from other studies and why?

4. The results section had many discussion points mixed in it. Please separate results from discussion. In the statistical analysis, why were t tests chosen? Could the balanced and imbalanced neurons have interaction effects that need to be handled in a different statistical model? How did your statistical model handle the "modality weight"? The differences in the vestibular-dominant and visual-dominant neuronal populations are no surprising, but what intrinsic anatomical and functional connections are there in the two systems to support the cross-modal integration in the MST area? What feature dimensions of cue disparity and reliability may influence the preferential coding strategies in decision-making? Please expand on the notion of modality dominance and the conditions such as input stability that may modulate the asymmetry in response decay between each input modality.

Reviewer 2 Report

This is a well written research article about an interesting topic, how visual-based spatial coordinate dominates probabilistic multisensory inference in disparity encoding of dorsal medial superior temporal area.

English language and style are generally fine, but there are some issues that need to be addressed before publication (e.g.  in lines 327-328 the word “functions” should be corrected to “function” and the references need to be modified according to the journal's reference style.

The introduction section describes the background of the current study in a comprehensive manner, mentioning not only the currents advances regarding neuronal correlates of sensory integration in the dorsal medial superior temporal area (MST-d), but also presenting the main key points of this study including assessing functional difference between the visual and vestibular signals during multisensory processing in MST-d and answer the question how the response distinction affects the decision making of integration or separation.

I think the methodology of this study should be described in a separate “materials and methods” section.

Results are quite interesting and well presented.

To my opinion, the discussion section is well written, discussing about the results of the current study and describing its strengths and limitations as well.

Perhaps, a separate conclusion section, summarizing the key points and giving some ideas for future research targets, would further improve the quality of the manuscript.

Round 2

Reviewer 1 Report

The revised manuscript showed a lot of improvement. I have one minor suggestion: The manuscript format needs to follow the standard for the journal.  Methods should be moved to Section 2 not 5. 

Author Response

Thanks to the reviewer for this point. We have fixed the order and placed methods section at the second section. We re-ordered the equation numbers and made some adjustments to the methods description about multisensory response curve averaging, please see line.207-216. Accordingly, we revised the reference of equation number in line.346, 362-363 in results section.